# Molecular Tools to Identify and Characterize Malignant Catarrhal Fever Viruses (MCFV) of Ruminants and Captive *Artiodactyla*

**DOI:** 10.3390/v14122697

**Published:** 2022-12-01

**Authors:** Laura Bianchessi, Mara Silvia Rocchi, Madeleine Maley, Renata Piccinini, Lauretta Turin

**Affiliations:** 1Department of Veterinary Medicine and Animal Sciences, University of Milan, 26900 Lodi, Italy; 2Moredun Research Institute, Pentlands Science Park, Bush Loan, Penicuik, Scotland EH26 0PZ, UK

**Keywords:** *Herpesviridae*, gamma(γ)-herpesvirus, malignant catarrhal fever virus (MCFV), consensus Panherpes PCR, long-distance PCR

## Abstract

The family *Herpesviridae* includes viruses identified in mammals, birds and reptiles. All herpesviruses share a similar structure, consisting of a large linear double-stranded DNA genome surrounded by a proteic icosahedral capsid further contained within a lipidic bilayer envelope. The continuous rise of genetic variability and the evolutionary selective pressure underlie the appearance and consolidation of novel viral strains. This applies also to several gamma(γ)-herpesviruses, whose role as primary pathogen has been often neglected and, among these to newly emerged viruses or virus variants responsible for the development of Malignant Catarrhal Fever (MCF) or MCF-like disease. The identification of γ-herpesviruses adapted to new zoological hosts requires specific molecular tools for detection and characterization. These viruses can cause MCF in livestock and wild animals, a disease generally sporadic but with serious welfare implications and which, in many cases, leads to death within a few days from the appearance of the clinical signs. In the absence of a vaccine, the first step to improve disease control is based on the improvement of molecular tools to identify and characterize these viruses, their phylogenetic relationships and evolutionary interaction with the host species. A Panherpes PCR-specific test, based on the conserved DNA polymerase gene, employing consensus/degenerate and deoxyinosine-substituted primers followed by sequencing, is still the preferred diagnostic test to confirm and characterize herpesviral infections. The drawback of this test is the amplification of a relatively short sequence, which makes phylogenetic analysis less stringent. Based on these diagnostic requirements, and with a specific focus on γ-herpesviruses, the present review aims to critically analyze the currently available methods to identify and characterize novel MCFV strains, to highlight advantages and drawbacks and to identify the gaps to be filled in order to address research priorities. Possible approaches for improving or further developing these molecular tools are also suggested.

## 1. Introduction

Malignant Catarrhal Fever (MCF), is a systemic typically fatal γ-herpesviral infection affecting domestic, captive and wild *Artiodactyla* worldwide. In absence of commercial vaccines or treatments available, and with limited diagnostic tests, MCF incidence is generally underestimated. The primary control method is based on careful management to prevent interaction between susceptible and non-susceptible hosts [1] in order to hinder virus transmission onto and within zoological species as well as the spread of the virus from reservoirs and carrier animals to susceptible species (e.g., from sheep to cattle). Despite the limited economic loss in domestic ruminants in Western Nations, wildebeest-associated MCF has substantial consequences in sub-Saharan Africa, where can account for up to 10% loss of cattle herds per year [2]. Furthermore, MCF may have a serious impact on the reduction of genetic diversity in the affected animal species, due to the possibility of severe outbreaks in zoological collections, which can affect already severely endangered animal species [3]. Finally, the knowledge of Malignant Catarrhal Fever Virus (MCFV) in terms of presence of new genetic variants, phylogenetic relationships and evolutionary link with the host species is also limited. Although some recent works attempted an analysis of Macaviruses genetic relationship [4,5] this was still based on the use of the Panherpes semi-nested PCR. An improvement in the genetic characterization of this viral genus could help the development of novel vaccines and better diagnostic tests, and may allow the identification of new adapted hosts that could act as vectors spreading the disease to non-adapted hosts. This is especially important for zoological species, where the knowledge about their specific herpesviruses is poor [6]. Due to the nature of the epidemiology of MCF, there are problems of interpretation of the diagnostic tests: in reservoir species antibody positivity is frequent, whereas PCR positivity is less frequent and can be intermittent in the same animal; conversely in affected species, PCR positivity is frequent in infected animals, whereas serological positivity is less frequent. Thus, the first step to improve disease control in livestock and zoological collections should be to improve molecular tools to identify and characterize the γ-herpesviruses that cause MCF. The currently used methods are based on consensus Panherpes PCR with consensus/degenerate and deoxyinosine-substituted primers and on long-distance PCR. The present review aims to describe and discuss these approaches, to identify the gaps and try to understand possible solutions based on techniques not yet applied to MCFV such as PCR with oligonucleotides modified by introduction of locked nucleic acids and next generation sequencing.

### 1.1. Herpesviruses

The family *Herpesviridae* includes viruses identified in mammals, birds and reptiles. The herpes virion is composed of a linear, double-stranded deoxyribonucleic acid (DNA) genome, surrounded by an icosahedral capsid, which is covered by a proteinaceous matrix (tegument), associated with an envelope containing glycoproteins (Figure 1) resulting from co-evolution and adaptation to the specific host. In the adapted hosts, these viruses establish life-long latent and subclinical infections. However, severe diseases may arise in the fetuses and newborn of the specific hosts and in immunocompromised or non-adapted hosts. In case of non-adapted hosts, such diseases generally occur when the viruses cross species barriers [7,8].

Based on biological and genetic characteristics and on molecular phylogenetic analyses, the family *Herpesviridae* is further divided into three subfamilies, *Alpha*(α)-*herpesvirinae*, *Beta*(β)-*herpesvirinae* and *Gamma*(γ)-*herpesvirinae*. These subfamilies arose nearly 200 million years ago and later evolved into sub-lineages probably through co-speciation with host lineages [9]. The viruses belonging to the subfamily γ-*herpesvirinae* have a restricted host range and establish life-long latent infection in mononuclear cells, mainly lymphocytes. This subfamily comprises seven genera, *Macavirus* (e.g., *Ovine* γ-*herpesvirus-2*), *Percavirus* (e.g., *Mustelid* γ-*herpesvirus-1*), *Lymphocryptovirus* (e.g., *Human* γ-*herpesvirus-4*), *Rhadinovirus* (e.g., *Human* γ-*herpesvirus-8*), *Bossavirus* (e.g., *Delphinind* γ-*herpesvirus-1*), *Manticavirus* (e.g., *Vombatid* γ-*herpesvirus-1*) and *Patagivirus* (e.g., *Vespertilionid* γ-*herpesvirus-3*) [10,11].

### 1.2. Malignant Catarrhal Fever Viruses and MCF

MCFVs belong to the genus *Macavirus*, which is the second (after *Rhadinovirus*) genus for member species number of the subfamily γ-*herpesvirinae*, and it is divided into two lineages of lymphotropic viruses [5], which include a group of viruses that may cause MCF (Table 1). In addition, several other similar herpesviruses, named based on the reservoir host from which they were identified can cause MCF, such as MCFV-white tailed deer and Ibex MCFV. MCFVs are antigenically and genetically related, being characterized by the presence of a 15-A antigenic peptide of the glycoprotein B and by high homology of the DNA polymerase sequence [5,12]. These viruses are adapted to their natural hosts, which are infected by their own MCFVs [5]. However, co-infections with more than one MCFV are possible [12,13]. MCF is a lymphoproliferative, often fatal disease. The main clinical signs are fever, profuse nasal discharge, ophthalmitis, corneal opacity, generalized lymphadenopathy, erosions of the upper respiratory tract and alimentary tract leukopenia. Diarrhea and neurologic signs may also occur. The affected animals die within a few days or up to several weeks after the beginning of the clinical signs. Several clinical outcomes may occur in different animal species depending on their degree of susceptibility to disease [14,15,16]. Clinical signs may also be species-specific.

Currently, only six MCFVs have been associated with clinical disease. The pathogenic viruses are *Alcelaphine* γ-*herpesvirus 1* (AlHV-1), *Ovine* γ-*herpesvirus 2* (OvHV-2), *Caprine* γ-*herpesvirus 2* (CpHV-2), *Ibex-MCFV*, *MCFV-white tailed deer* (WTD)/*Caprine* γ-*herpesvirus 3* (CpHV-3), and *Alcelaphine* γ-*herpesvirus 2* (AlHV-2). These viruses infect many ungulates species of the order *Artiodactyla*. MCF can occur in non-adapted hosts that share enclosures or are co-grazed with the adapted hosts, which carry their own MCF viruses latently or subclinically; the virus is secreted in their ocular and nasal discharge and can be transmitted directly to a susceptible host or contaminate the environment [14]. Currently, the most studied MCFVs are AlHV-1 and OvHV-2, the only viruses with a complete sequenced genome [6]. Consequently, the epidemiology of MCF is well defined only for these two MCFVs [14]. AlHV-1 is carried by blue and black wildebeest and causes wildebeest-associated MCF (WA-MCF) especially in cattle and deer. WA-MCF occurs most frequently in Africa during wildebeest calving season, although sporadic cases have been reported by zoological parks or game farms worldwide [14,20,21]. On the contrary, OvHV-2 is carried by domestic and wild sheep and causes sheep-associated MCF (SA-MCF) mainly in cattle, deer, bison and occasionally in pig and giraffe. SA-MCF is present worldwide [5,14]. For example, in the UK is common to test sheep for OvHV-2 and wildebeest for AlHV-1, before introduction or movement between zoological collections. This however does not take in account of the potential presence of other MCFV in these and other species which could be responsible for disease outbreaks.

Due to the above-mentioned diagnostic challenges, in Europe and in the UK, there are very few data regarding the molecular epidemiology of the disease in livestock and zoological collections. Thus, new diagnostic tests are needed to improve surveillance and characterization and to define the source of outbreaks [14,21,22].

## 2. Molecular Methodologies

Several methodologies have been developed to aid the detection and identification of MCFV, and to allow phylogenetic characterization. In the section below are presented the different techniques used, discussing their relevance to MCFV as well as presenting prospective new approaches to overcome the limitations of the more classical techniques initially developed.

### 2.1. Consensus Polymerase Chain Reaction Assays

The Consensus Polymerase Chain Reaction (PCR) assay is a tool that can be used to amplify sequences of orthologue genes from related organisms, also allowing the identification of novel pathogen species or strains via phylogenetic analysis. Primers are designed from highly conserved regions present in orthologous sequences, which are first identified by multiple alignment of known sequences evolutionarily related to the targeted orthologous. The primers design is based on two strategies [23,24]. The first strategy is to synthesize a pool of different oligonucleotide sequences, called degenerate primers, based on known nucleotide or amino-acid sequences of highly conserved regions, with each primer containing a possible nucleotide variant of the conserved nucleotide region [23,25]. It is generally not advisable to design degenerate primers using low conserved regions since this would result in a disproportionate low yield of specific product in the final PCR reaction and a high yield of unspecific products. Moreover, primers with different AT/CG nucleotide content or length may cause problems in finding the right annealing temperature, also reducing sensitivity and specificity [24,25]. The only advantage of consensus PCR with degenerate primers is the possibility to amplify sequences from unknown and distantly related viruses. The second strategy is to design a single sequence, called consensus primer which contains the most common nucleotide at each codon position in the conserved regions. Such a primer allows the identification of highly conserved homologous genes, but will miss less related sequences [24,25] or novel viruses. Consequently, PCRs based on degenerate or consensus primer strategies are useful for specific applications, such as to detect highly related genes which are present in adequate concentrations, but they show low specificity and sensitivity when used to detect distantly related genes or genetic material present at low abundance [24].

### 2.2. Consensus Degenerate Hybrid Oligonucleotide Primers PCR Assays

Consensus PCR assays can be improved by combining the characteristics of the two primers, consensus and degenerate, as described by Timothy M. Rose et al. These primers are called Consensus-Degenerate Hybrid Oligonucleotide Primers (CODEHOPs) [25]. CODEHOPs are a pool of primers where each oligonucleotide is composed of a short 3′ degenerate core region and a long 5′ consensus clamp region [25,26,27]. These primers are designed based on the nucleotide sequences coding for conserved amino-acid regions, which are identified by multiple alignment of known protein sequences evolutionarily related to the targets. These conserved regions contain a motif of 3 or 4 highly conserved amino-acids that are encoded by slightly degenerated nucleotide codons, flanked upstream (forward primer) or downstream (reverse primer) by nucleotide sequences coding for conserved amino-acid residues [25]. Therefore, the short 3′ degenerate core region will contain all possible codons encoding highly conserved amino-acid motifs, while the long 5′ consensus clamp region will contain the most common nucleotides encoding conserved amino-acid residues. This allows to reduce the number of individual primers employed and, at the same time, increases the concentration of the more specific primers in the pool [25,28]. Consequently, the specificity of the reaction increases during the initial PCR amplification, when the primers containing the correct short 3′ degenerate core region hybridize without any mismatch to the target DNA template and this hybridization is stabilized by long 5′ consensus clamp region, which allows the use of higher annealing temperature. The early cycles of PCR will generate a high yield of specific products containing the primer sequences; in the subsequent cycles, the hybridization between primers and specific products will be driven by the long 5′ consensus clamp region [25,26,28]. Therefore, a CODEHOPs strategy increases the specificity and sensitivity of this PCR assay to detect and amplify distantly related genes encoding conserved amino-acid regions, in complex mixtures of genetic material of different pathogen species [24,25,26].

The consensus PCR has been employed to identify potential unknown herpesviruses also based on protein domains [29] since this family is characterized by synonymous nucleotide substitutions being more numerous than non-synonymous substitutions [30]. Consequently, a core set of orthologous genes had been identified, containing sequences encoding amino-acid domains conserved in all mammalian herpesviruses including genes encoding the major DNA binding protein, the DNA-packaging gene, the catalytic subunit of DNA polymerase, the glycoprotein B (gB), the major capsid protein and the DNA helicase and uracil-DNA glycosylase genes. For most of these genes, complete nucleotide or amino-acid sequences are available for a range of virus species [9]. Thus, the phylogenetic analysis is carried out by aligning sufficiently conserved amino-acid sequences encoded from these genes [9,31].

### 2.3. Consensus Polymerase Chain Reaction Assay for Herpesvirus Detection

Conserved amino-acid regions within the DNA polymerase sequences were originally identified in 1991 by Ito and Braithwaite by alignment of 40 different DNA polymerase sequences [32]. VanDevanter et al. used three of these conserved amino-acid regions (A, B and C) to design a combination of degenerated and CODEHOP PCR primers (DFA, ILF, TGV, IYG and KG1) able to amplify a short (215- to 315-bp) nucleotide sequence located between the regions B and C of the herpesvirus DNA polymerase [33]. The DFA primer is completely degenerated, while ILF, TGV, IYG and KG1 primers were designed using the manual CODEHOP strategy [28]. The primers are used in a nested format with the primary PCR being performed with two forward (DFA and ILK) and one reverse (KG1) primer. The secondary PCR is performed with one forward (TGV) and one reverse primer IYG (Table 2a). Secondary PCR products can be sequenced directly by Sanger method, using primers TGVseq and IYGseq (Table 2a) or the same primers used in the secondary PCR [33]. The advantage of these universal primers is that they have the capability to detect virtually any mammalian, avian or reptilian herpesvirus (Figure 2).

Template dilution studies revealed a good sensitivity of this method [31,33]. Therefore, such consensus PCR represents a powerful tool to identify and characterize both known and unknown herpesviruses in human and animal samples. This was demonstrated by VanDevanteret et al., who detected and characterized 7 known herpesvirus species (5 humans and 2 animals) and 14 unknown herpesvirus species (2 humans and 12 animals) [33]. Other researchers identified and characterized new *γ-herpesviruses* present in *Artiodactyla* species by using this approach [29,34,35,36]. Analysis of the DNA polymerase amino-acid sequence between the conserved regions B and C shows that this region is unique in different herpesvirus species [33] and that the corresponding phylogenetic trees correlate with the biological subfamily classification of herpesviruses.

Nevertheless, the specificity and sensitivity of this consensus PCR is not optimal when investigating all herpesvirus species, because the amino-acid motifs used to derive the nucleotide sequence used to design primers are not conserved in all herpesvirus species and are encoded by highly degenerated codons, leading to the design of primers unable to bind with the same affinity to DNA polymerase genes from different herpesvirus species. High degeneracy can also lead to an increase in the number of primers in the pool, with lack of amplification of the DNA polymerase genes in tissue samples with low abundance of viral genome since sensitivity is inversely correlated to the number of primers present [28].

### 2.4. Consensus Panherpes PCR with Consensus/Degenerate and Deoxyinosine-Substituted Primers

The original consensus PCR described above was eventually improved by Ehlers et al., by using deoxyinosine-substituted primers [37] and introducing deoxyinosine in all primer positions with 3- and 4-fold degeneration (Figure 3) [37]. Deoxyinosine (I^a^) is a universal base that can pair with all four nucleotides, without destabilizing the bond. Thus, deoxyinosine residues may be introduced in primer positions of maximum degeneracy to decrease the complexity of the degenerate primer pool. This generates a higher yield of specific PCR products. [38]. However, deoxyinosine-substituted primers performance depends on the ratio of I residues and primer length, and the nature of the template, limiting the amplification of the consensus sequences of all virus species, despite modified reaction conditions [38].

Subsequently, the same authors, using a mixture of degenerate/consensus and deoxyinosine-substituted primers (Table 2b) in a nested format identified and characterized 16 known and 3 unknown animal herpesvirus species. This PCR based on deoxyinosine-substituted and degenerate/consensus primers, called consensus Panherpes PCR, is a powerful tool to universally detect unknown mammalian, avian and reptilian herpesviruses, including MCFVs [37]. Two novel porcine γ-herpesviruses were discovered by using this method, porcine lymphotropic γ-herpesvirus type 1 and type 2 (PLHV-1 and PLHV-2), respectively [39]. However, even this improved PCR shows limitations. First, it amplifies a short sequence of approximately 215- to 315-nucleotides depending on which primers are successful. Therefore, precise phylogenetic analysis may be limited, because short sequences are not sufficient to construct phylogenetic trees revealing acceptable probabilities for all clades. Short sequences are also usually not submitted to repository databases [40], limiting the number of sequences the amplicon can be compared to and leading to inaccurate identification since mismatches could either be attributed to actual sequence evolution or PCR errors. The second shortcoming is that the Panherpes consensus PCR may not amplify all herpesvirus strains present in the same sample if one or more strains are more abundant than others. This was demonstrated by the discovery of *Porcine lymphotropic γ-herpesvirus 3* (PLHV-3) in porcine blood and tissue samples previously confirmed positive only (PLHV-1 and PLHV-2) by Panherpes consensus PCR [39,40,41]. Thus, the Panherpes consensus PCR may not detect unknown herpesviruses present at low abundance in samples infected with more than one virus [40] due to template competition issues and the exponential nature of the PCR.

### 2.5. Ovine Herpesvirus 2 Sequence Analysis 

A multi-locus approach, based on a combination of three conventional nested PCR assays, was used by Russell et al. [42] to analyze the genetic diversity of OvHV-2. The genes analyzed represented polymorphic *loci* (segments of *ORF50*, *ORF75* and *Ov9.5*, the latter also present in AlHV-1). Starting from cattle diagnostic samples representing MCF cases with different clinical presentations, three genes of OvHV-2 were amplified from each sample and Sanger sequenced utilizing the internal PCR primers as the sequencing primers. This led to the sequencing of, respectively, 444, 238 and 893 bp products which were subsequently subjected to combined phylogenetic and evolutionary analysis using a maximum likelihood approach and the Tamura-Nei model. This analysis showed twelve distinct viral variants, with the clades distribution affected by the most polymorphic *locus* (*Ov9.5*) but still influenced by further differences detected in the fragments of *ORF50* and *ORF75* genes. Although no association was noted between specific variants of OvHV-2 identified and clinical presentation, this high-resolution multi-locus approach shows the potentiality for it use in epidemiological studies and in the establishment of outbreaks chain of infections.

### 2.6. Long-Distance Polymerase Chain Reaction Assay and Primer Walking

Chmielewicz et al. discovered CpHV-2, a MCFV housed and spread from goats, with a bigenic approach employing a so-called long-distance PCR. This PCR assay connects the short PCR-derived segments of the glycoprotein B gene (*gB*) with the DNA polymerase gene and produces a final contiguous sequence about 3.6 kbp [43]. The glycoprotein B gene in β- and γ-herpesvirus genomes is adjacent and immediately upstream of the DNA polymerase gene. However, the *gB* gene is less conserved than the DNA polymerase gene and is only conserved among herpesviruses belonging to the same subfamily or genus [40]. Consequently, in order to detect the glycoprotein B genes is necessary to design more specific degenerate or consensus primers, which will amplify sequences of a single herpesvirus subfamily or genus. Thus, this PCR assay might require several attempts with different primers to obtain the long sequence [40].

Two parallel consensus PCRs are used to obtain the long sequence: the Panherpesvirus consensus PCR for the DNA polymerase gene [37] and a consensus PCR for the glycoprotein B gene, where degenerate and deoxyinosine-substituted genus specific primers are frequently used in a nested or semi-nested format [40,43]. The two sequences originated from the initial PCR assays are then connected by a nested mid-distance PCR employing forward glycoprotein B specific primers and reverse DNA polymerase specific primers [40,43,44]. Finally, long contiguous PCR products are sequenced though primer walking with the corresponding amino-acid sequences predicted by bioinformatics software. Primer walking is an iterative method which uses a starting primer to obtain an initial short sequence via Sanger sequencing, followed by the design of a new primer, located near the end of the initial sequence, which is then used to sequence further bases thus “walking” along the sequence [45]. For example, Chmielewicz et al. used three degenerate and deoxyinosine-substituted primers in a semi-nested PCR format to achieve glycoprotein B gene detection as shown in Table 3a. These primers were designed based on the glycoprotein B nucleotide sequences of AlHV-1 and only amplify sequences of this and related viruses [43]. Using primer walking and a log PCR Ehlers and collaborators identified 14 novel gamma-herpesviruses in specimens from hosts from six mammalian orders of which 8 were previously unknown [46]. In particular, the glycoprotein B gene was amplified with two primer sets (GH1 and GH2) of degenerate/consensus primers and deoxyinosine-substituted genus specific primers in a nested PCR format. The primary and nested PCRs were performed by employing pools degenerate/consensus and deoxyinosine-substituted primers, as detailed in Table 3b. This study demonstrated for the first time that it is possible to develop a mid-distance PCR for herpesviruses. A limited number of degenerate/consensus primers and deoxyinosine-substituted primers is supposed to be sufficient for the universal detection of mammalian γ-herpesviruses [46].

This work demonstrated that robust phylogenetic analysis (both at nucleotide and amino-acid level) can be achieved by using long contiguous sequences, therefore overcoming one of the problems of the Panherpes consensus PCR, as demonstrated by the generation of phylogenetic trees with significantly high probability [46]. Thus, mid-to-long-distance PCRs could be a successful strategy to achieve better identification and characterization of MCFVs. Nevertheless, its use is restricted and limited by the specificity of the primers used, which will not amplify all members of the *Herpesviridae* family.

### 2.7. Locked Nucleic Acids Approaches

When the *gB* sequence and the DNA polymerase sequence do not originate from the same virus genome, for example during mixed infection, the utility of the previously mentioned approaches is limited [46]. This can be resolved using primers containing Locked Nucleic Acid (LNA). LNA refers to nucleotides designed using a modified RNA analogue base which shows increased stability against enzymatic degradation. LNA oligonucleotides hybridize with high affinity with complementary double-stranded DNA and can be mixed with DNA or RNA residues in a oligonucleotide. The wide applicability of LNA oligonucleotides for gene silencing and their use for research and diagnostic purposes are documented in several recent reports [47,48,49].

LNA technology can be applied to γ-herpesvirus detection and characterization by carrying out consensus PCRs in the presence of LNA oligonucleotides, which specifically inhibit the PCR amplification of herpesvirus known to be present in the sample, however still allowing the amplification of under-represented target herpesvirus sequences. This approach, in combination with a bigenic *DNApol/gB* amplification, was used by Prepens et al. to analyze multivirus-infected chimpanzees and macaques and identified six new herpesviruses of the genera *Rhadinovirus*, *Lymphocryptovirus* and *Cytomegalovirus* [40]. More specifically, one of the oligonucleotides usually employed in the consensus Panherpes *DNApol* PCR (see Section 2.4) is substituted with a LNA oligonucleotide. This blocks the amplification of the known-to-be present herpesvirus DNApol sequence and allows the amplification of a second, less represented *DNApol* sequence. This allows the identification of the corresponding, less conserved, *gB* gene which is then targeted with several sets of degenerate primers. The two sequences obtained using this approach can then be connected by a long-distance PCR. Therefore, this technique allows the amplification of longer sequences of multiple viruses from a single specimen. In particular, Panherpes PCR in presence of a multiple sequence-specific LNAs allows the exclusion of undesired viruses amplification, enabling the amplification of other, possibly novel viruses.

### 2.8. Next Generation Sequencing

Next-generation sequencing (NGS) is becoming a prominent technology for viral identification and diagnosis, allowing the detection of known, unknown and potentially emerging viruses in the same sample. This sequencing approach has the potential to reduce the number of assays required for virus identification and can provide fast full-length viral genome sequences, improving the speed and the accuracy of the diagnostic process. NGS, despite having been recently employed for testing livestock samples for virus discovery [50] has still not been applied to MCFV, notably because of some common drawbacks. 

Initial amplification of the viral DNA seems to be an important step in obtaining appropriate samples for sequencing. For example, whole genome sequencing of a bovine Herpesvirus 4 strain was achieved from MDBK-infected cells via cesium chloride purification and Roche 454 FLX by Gagnon et al. [51], and similarly Kaposi sarcoma herpesvirus samples were initially enriched by patient’s therapeutic or diagnostic thoracentesis, pericardiocentesis, or paracentesis [52] followed by Illumina sequencing. Human Epstein–Barr virus was directly sequenced from nasopharyngeal carcinoma tissue of a patient, and its sequence derived via Illumina genome analysis [53] using whole genome EBV wild-type sequences as reference, after removal of human genome sequences. Additional herpes viruses were also recently identified by mining livestock genome, including Bovine γ-HV-4 and 6, suid γ-herpesvirus 3 and 4 and gallid α-herpesviruses 1, 2 and 3 [54]. However, in all these cases, host reference genomes were available for subtraction and viral sequences could be compared to sequences deposited in the NCBI Viruses resource.

The lack of permissive cell culture system for MCFV to amplify the genomes present in the starting material does not allow the preparation of sufficient good quality DNA for the analysis. The complexity of sample processing for NGS, the cost, and the sophisticated computational resources required for assembly, annotation and analysis of its massive amounts of outputs are additional barriers that need to be addressed. Genome sequencing with nanopore technology of large viral genomes, such as the one of *Herpesviridae* family, can be difficult due to the size, the high GC content and the replication-derived nucleotide variations. Moreover, the pipelines allowing host genome subtraction for many of the target species (either carrier or affected) are not yet available. More strategies would be required in order to increase the efficiency, reduce the sequencing errors, maximize the reproducibility and ensure a correct data management when approaching these technologies.

## 3. Conclusions and Future Perspectives

The most relevant molecular tool used so far to identify and characterize the γ-herpesviruses causing MCF in livestock and zoological species is the Consensus Panherpesvirus PCR. This PCR uses primers that amplify the DNA polymerase gene of all members of *Herpesviridae* family with the advantage of universality. The disadvantage is the small size of the amplified region, which may lead to inaccurate identification or less precise phylogenetic analysis, although multigenic, statistical and bioinformatic methodologies which would allow better classification have been published [50].

In recent years, in order to overcome this shortcoming, researchers have used mid-to-long-distance PCR approaches, which connect the glycoprotein B gene with the DNA polymerase gene to produce a final contiguous sequence of approximately 3.6 kbp. This type of PCR allows more accurate identifications and improved phylogenetic analyses of the virus although it requires *gB* gene-specific genus primers and can be lengthy and costly. By using γ-herpesvirus universal primers, this approach could be applied to identify and characterize viruses belonging to the genus Macavirus. This approach, although potentially in common within each genus, as well as efficient and specific, could give false negative results for samples with low virus abundance, being less sensitive than the Consensus Panherpes PCR, and for samples latently infected by more than one herpesvirus. Recent or newer approaches such as the use of Locked Nucleic acids or Next Generation Sequencing can be promising and constant technology development might soon allow their use for the detection and characterization of MCFV.

Additional studies are necessary to validate molecular tools to detect and characterize the viruses that cause MCF. These will provide information about new genetic variants, phylogenetic relationships and evolutionary links, and consequently will help to develop vaccines and diagnostic tests, improving the surveillance system.

## Figures and Tables

**Figure 1 viruses-14-02697-f001:**
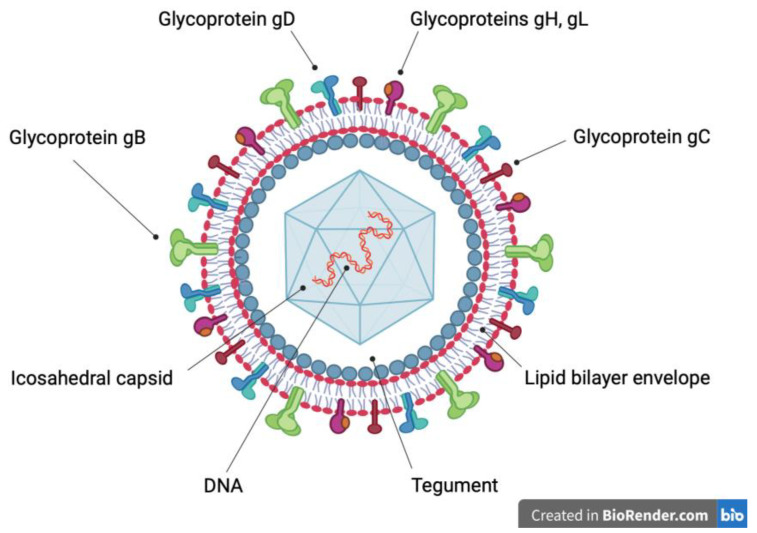
Schematic representation of the herpes virion. A linear, double-strand deoxyribonucleic acid (DNA) molecule (red and orange) is surrounded by an icosahedral capsid (light blue), which in turn is covered by tegument (white). The outer surface of the tegument (grey) is associated with a lipid bilayer envelope (red and grey), which contains integral glycoproteins. Glycoprotein B trimer (green), glycoprotein C monomer (brown), glycoprotein D homodimer (light blue), glycoprotein H (purple) and glycoprotein L (orange) heterodimer allow the entry of the virion into the host cell.

**Figure 2 viruses-14-02697-f002:**
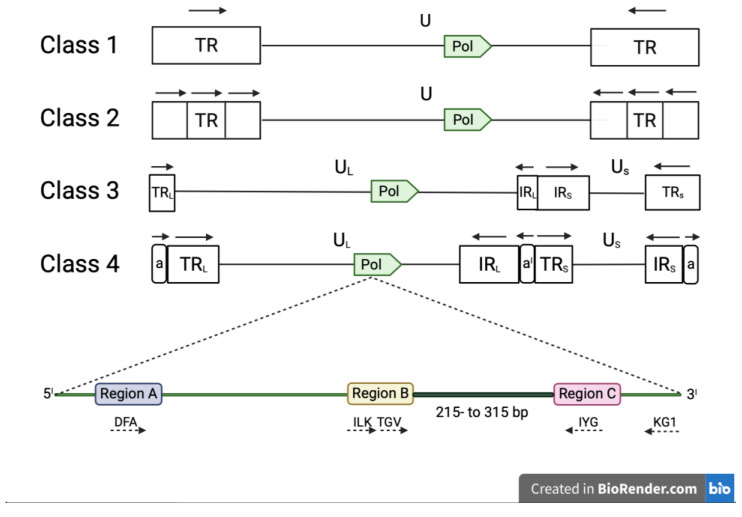
Classes of viruses belonging to the family *Hespesviridae* based on the arrangement of repeated sequences in the genome. *Human herpesvirus-6A* and *Human herpesvirus-6B* (subfamily β-*herpesvirinae*) and *Equine herpesvirus-2* (subfamily γ-*herpesvirinae*) genomes belong to class 1, while many members of the subfamily γ-*herpesvirinae* belong to class 2; members of the genus Varicellavirus (subfamily α-*herpesvirinae*) belong to class 3 genome, whereas Herpes simplex virus (subfamily α-*herpesvirinae*) and Cytomegalovirus (subfamily α-*herpesvirinae*) belong to class 4. The *pol* gene, which contains three conserved regions (A, B and C), is present in the architecture of all genome classes (illustrations not at scale). These regions encode highly conserved amino acid domains (DFA, ILK, TGV, IYG and KG1). PCR primers (dashed arrows) according to VanDevanter et al. are shown under the different class architectures. Primer’s directions are specified by dashed arrows. The product obtained by the Panherpes nested PCR is between region B and region C. The nomenclature used in all classes is U (unique), U_L_ (unique long), U_S_ (unique short), TR (terminal repeat), TR_L_ (terminal long repeat), IR_L_ (internal long repeat), TR_S_ (terminal short repeat), IR_S_ (internal short repeat); TR_L_ and TR_S_ include a (terminal direct repeat) and IR_L_ and IR_S_ a^I^ (internal inverse repeat). The orientations of repeated sequences are specified by block arrows.

**Figure 3 viruses-14-02697-f003:**
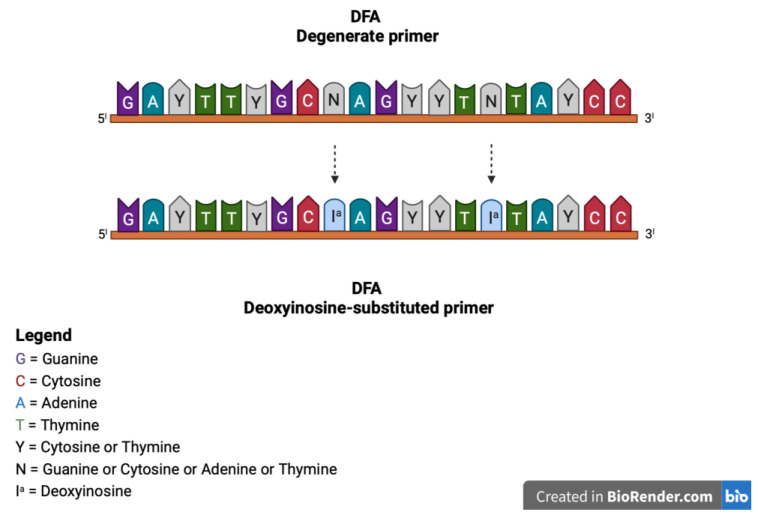
Representation of the deoxyinosine-substituted primers generated by Ehlers et al. modifying the original degenerate primers and introducing deoxyinosine in all primer positions with 3- and 4-fold degeneration. For example, in the DFA degenerate primer, the degenerate N base was substituted with I^a^.

**Table 1 viruses-14-02697-t001:** Macaviruses (shaded) and non Macavirus Malignant Catarrhal Fever Viruses, their clinical significance, reservoir(s) and susceptible species. Based on the ICTV classification at time of printing.

Virus	Clinical MCF	Reservoir Species	Susceptible Species	Reference
*Alcelaphine* γ-* herpesvirus-1*	Yes	Wildebeest	Cattle Deer	[14]
*Alcelaphine* γ-* herpesvirus-2*	Yes	Hartebeest Topi	Barbary red deer Bison	[14]
*Ovine* γ-*herpesvirus-2*	Yes	Sheep Mouflon	Cattle Water buffalo Moose Domestic goats Deer Bison Pig Giraffe	[1,14]
*Caprine* γ-*herpesvirus-2*	Yes	Goat	Sika deer White-tailed deer	[14]
*Hippotragine* γ-*herpesvirus-1*	No	Roan antelope	No susceptible species identified	[14]
*Bovine* γ-*herpesvirus-6*	No	Cattle	No susceptible species identified	[17]
*Suid* γ-*herpesvirus-3*	No	Pig	No susceptible species identified	[18]
*Suid* γ-*herpesvirus-4*	No	Pig	No susceptible species identified	[18]
*Suid* γ-*herpesvirus-5*	No	Pig	No susceptible species identified	[18]
MCFV–white-tailed deer/*Caprine* γ-*herpesvirus-3*	Yes	Goat	White-tailed deer Red brocket deer Reindeer	[14]
Ibex-MCFV	Yes	Ibex	Bongo antelopes Anoa Pronghorn Duikers	[14,19]

**Table 2 viruses-14-02697-t002:** Primers sequences employed for the Degenerate/Consensus PCR according VanDevanter et al. (**2a**) and the Consensus Panherpes PCR according to Ehlers et al. (**2b**). Primers used for Sanger Sequencing following the Degenerate/Consensus PCR according to VanDevanter et al. are shown in italic. Degenerate and inosine-substituted equivalents are shown in bold. I^a^ = Deoxyinosine.

2a. Consensus Panherpes Polymerase Chain Reaction Primers	
** First PCR **	
** *Forward* **	** *Reverse* **
**DFA**: 5′-GA**Y** TT**Y** GC**N** AG**Y** YT**N** TA**Y** CC-3′	**KG1**: 5′-GTC TTG CTC ACC AG**N** TC**N** AC**N** CC**Y** TT-3′
**ILK**: 5′-TCC TGG ACA AGC AGC AR**N Y**SG C**NM** T**N**A A-3I’	
** Second PCR **	
** *Forward* **	** *Reverse* **
**TGV**: 5′-TGT AAC TCG GTG TA**Y** GG**N** TT**Y** AC**N** GG**N** GT-3′	**IYG**: 5′-CAC AGA GTC CGT **R**TC **N**CC **R**TA DAT-3′.
***Approximate product****:* 215- to 315-bp	
**Sanger Sequencing**	
** *Forward* **	** *Reverse* **
*TGVseq: 5′-CAT CTG ATG TAA CTC GGT GTA-3′ bottom line*	*IYGseq: 5′-GAC AAA CAC AGA GTC CGT-3′*
**2b. Consensus Panherpes PCR with Consensus/Degenerate** **and Deoxyinosine-Substituted Primers**	
** First PCR **	
** *Forward* **	** *Reverse* **
**DFA**: 5′-GAY TTY GC**N** AGY YT**N** TAY CC-3′	**KG1**: 5′-GTC TTG CTC ACC AG**N** TC**N** AC**N** CCY TT-3′
**ILK**: 5′-TCC TGG ACA AGC AGC AR**N** YSG C**N**M T**N**A A-3′	
**Deoxyinosine-substituted equivalent**	**Deoxyinosine-substituted equivalent**
**DFA**: 5′-GAY TTY GC**I^a^** AGY YT**I^a^** TAY CC-3^I^′	**KG1**: 5′-GTC TTG CTC ACC AG**I^a^** TC**I^a^** AC**I^a^** CCY TT-3′
**ILK**: 5′-TCC TGG ACA AGC AGC ARI**^a^** YSG CI**^a^**M TI**^a^**A-3′	
** Second PCR **	
** *Forward* **	** *Reverse* **
**TGV**: 5′-TGT AAC TCG GTG TAY GG**N** TTY AC**N** GG**N** GT-3′	**IYG**: 5′-CAC AGA GTC CGT RTC **N**CC RTA **D**AT-3′
**Deoxyinosine-substituted equivalent** **TGV**: 5′-TGT AAC TCG GTG TAY GG**I^a^** TTY AC**I^a^** GG**I^a^** GT-3′	**Deoxyinosine-substituted equivalent** **IYG**: 5′-CAC AGA GTC CGT RTC **I^a^**CC RTA **I^a^**AT-3′

**Table 3 viruses-14-02697-t003:** Primer sequences for the Semi nested consensus PCR for the amplification of fragment of the glycoprotein B according to Chmielewicz et al. (**3a**) and Degenerate/Consensus primers and Deoxyinosine-substituted primers used by Ehlers et al. in the gB nested PCR (**3b**). I^a^ = Deoxyinosine.

3a. Semi-Nested PCR for the Glycoprotein B (gB) Gene	
** First PCR **	
** *Forward* **	** *Reverse* **
**702 Gb**: 5′-CAR I^a^TI^a^ CAR TWT GCM TAY GAC-3′	**702 Gb**: 5′-GTA RTA RTT RTA YTC YCT RAA-3′
** Second PCR **	
** *Forward* **	** *Reverse* **
**734** ***gB***: 5′-GCA AAA TCA ACC CTA CVA GYG TNA TG-3′	**702** ***gB***: 5′-GTA RTA RTT RTA YTC YCT RAA-3′
***Approximate product****:* 515bp	
**3b. Nested PCR for the Glycoprotein B (gB) Gene**	
** First PCR **	
** *Forward* **	** *Reverse* **
**GH1 2759**: 5′-CCT CCC AGG TTC ART WYG CMT AYG A-3^I^	**GH1 2762**: 5′-CCG TTG AGG TTC TGA GTG TAR TAR TTR TAY TC-3′
**GH2 3029**: 5′-CCC AGT TGC ART WYG GC(N/I^a^) TAY GA-3′	**GH2 3033**: 5′-GCC AGG CGT TGC GT(N/I^a^) TAR TAR TTR TA-3′
** Second PCR **	
** *Forward* **	** *Reverse* **
**GH1 2760**: 5′-AAG ATC AAC CCC AC(N/I^a^) AG(N/I^a^) GT(N/I^a^) ATG-3′	**GH1 2761**: 5′-GTG TAG TAG TTG TAC TCC CTR AAC AT(N/I^a^) GTY TC-3′
**GH2 3031**: 5′-CAA GAT TAA CCC CAC (N/I^a^)AG (N/I^a^)GT (N/I^a^)AT G-3′	**GH2 3032**: 5′-TTG CGT GTA GTA GTT GTA YTC (N/I^a^)CT RAA CAT-3′

***Approximate product GH1***: 500 bp; ***Approximate product GH2***: 350 bp.

## Data Availability

Not applicable.

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
