# Peer review of "Molecular Tools to Identify and Characterize Malignant Catarrhal Fever Viruses (MCFV) of Ruminants and Captive Artiodactyla"

_viruses, 2022, doi:10.3390/v14122697_

Round 1
Reviewer 1 Report
The manuscript by Bianchessi et al. reviews the current published methods that have been developed to identify circulating strains of macavirus members; by using consensus/degenerate PCR approaches. The authors describe in details the principles of using degenerate primers to target consensus conserved regions on herpesvirus genomes, mostly encoding the viral DNA polymerase gene, a highly conserved gene in its peptidic sequence.
Although the manuscript is of interest for an audience unaware of the consensus PCR strategies, the level of details is at some places too high while referring to only very few papers that are mostly orientated on MCFV detection and identification of new variants (one report of the pan-herpes PCR by VanDevanter et al in 1996 and some papers from Ehlers et al.).
It would have been much interesting if this manuscript would actually compare these PCR-based methods with other alternative approaches that would potentially allow a more robust manner of identifying larger sequences of viral genomes to perform phylogenetic analyses downstream. Although LNA-primers, and primer walking were introduced, the authors do not provide unpublished alternatives that could improve the current developed tools, either in the context of research or in the context of diagnosis in the field. For example, would new nextgen sequencing technologies on materials in which a consensus PCR as turned positive would be of interest? What do the authors think about new developments? These would be concepts and perspectives that one might like to read in such a mixed MCFV and molecular biology-orientated review.
Minor comments:
- There are still typological errors (a gamma is missing in the abstract, herpesvirus for AlHV-1 in Table 1, …)
- Line 74: ‘antigenic epitope’: perfer antigenic peptide or epitope as epitope refers on its own to a some part of an antigen.
- Line 76: ‘to their own hosts’: prefer ‘to their natural hosts’ instead.
- Table 1 was made based on the Li et al. (ref 9); and could have been more developed. I wouldn’t think AlHV-1-induced MCF only occurs in cattle and deer.
- Tables with primer sequences: there should be an alignment with the different rows ion each columns. These tables should be useful as they are meant to contain all sequence information about the primers to be used if researchers want to use these PCRs. The authors should pay much attention to these tables and their format to ease their use.
- Lines 184-189: the fact that the primers are designed based on the conserved peptidic sequence renders the explanation ambiguous. This should be clarified as it reads like if the primers contain amino acid residues, which does not make sense.
- There are twice the same title Section 2.1. This should be clarified.
- Section 2.2.2 exists without section 2.2.1
Author Response
Point-by-point Response (in red) to Reviewer (in black)
Reviewer reports:
Reviewer #1: Comments and Suggestions for Authors:
The manuscript by Bianchessi et al. reviews the current published methods that have been developed to identify circulating strains of macavirus members; by using consensus/degenerate PCR approaches. The authors describe in details the principles of using degenerate primers to target consensus conserved regions on herpesvirus genomes, mostly encoding the viral DNA polymerase gene, a highly conserved gene in its peptidic sequence.
Although the manuscript is of interest for an audience unaware of the consensus PCR strategies, the level of details is at some places too high while referring to only very few papers that are mostly orientated on MCFV detection and identification of new variants (one report of the pan-herpes PCR by VanDevanter et al in 1996 and some papers from Ehlers et al.).
It would have been much interesting if this manuscript would actually compare these PCR-based methods with other alternative approaches that would potentially allow a more robust manner of identifying larger sequences of viral genomes to perform phylogenetic analyses downstream. Although LNA-primers, and primer walking were introduced, the authors do not provide unpublished alternatives that could improve the current developed tools, either in the context of research or in the context of diagnosis in the field. For example, would new nextgen sequencing technologies on materials in which a consensus PCR as turned positive would be of interest? What do the authors think about new developments? These would be concepts and perspectives that one might like to read in such a mixed MCFV and molecular biology-orientated review.
We extensively revised the manuscript according to the helpful suggestions provided, including also more references. The articles cited (now more), albeit not many and somehow dated, are still the only available references for the molecular methods applied to identify and characterize MCFV, the topic of the present review. The high level of details is justified, in our view, by the complexity of the design of consensus PCR assays suitable to detect phylogenetically related viruses characterized by considerable variability. We agree with the referee’s comment, so we to compared the described PCR methods with alternative and more recent approaches.
Minor Comments:
- There are still typological errors (a gamma is missing in the abstract, herpesvirus for AlHV-1 in Table 1, …)
We corrected this and other English language mistakes or words inconsistently written throughout the text.
- Line 74: ‘antigenic epitope’: perfer antigenic peptide or epitope as epitope refers on its own to a some part of an antigen.
We corrected as suggested by replacing “epitope” with “peptide“ (previous line 74, now line 108).
- Line 76: ‘to their own hosts’: prefer ‘to their natural hosts’ instead.
We corrected as advised (previous line 76, now line 109).
- Table 1 was made based on the Li et al. (ref 9); and could have been more developed. I wouldn’t think AlHV-1-induced MCF only occurs in cattle and deer.
We updated Table 1 by adding few animals susceptible to Ovine g-herpesvirus-2 and an animal (from a recent published paper) susceptible to Ibex-MCFV with relative references.
- Tables with primer sequences: there should be an alignment with the different rows ion each columns. These tables should be useful as they are meant to contain all sequence information about the primers to be used if researchers want to use these PCRs. The authors should pay much attention to these tables and their format to ease their use.
We aligned rows and columns in Tables 2 and 3 as requested and according to the Journal format.
- Lines 184-189: the fact that the primers are designed based on the conserved peptidic sequence renders the explanation ambiguous. This should be clarified as it reads like if the primers contain amino acid residues, which does not make sense.
We rephrased the sentence at previous lines 184-189 (now lines 185-191) in order to clarify the ambiguity.
- There are twice the same title Section 2.1. This should be clarified.
- Section 2.2.2 exists without section 2.2.1
In response to the above 2 points, we reviewed the style, we corrected the titles and the numbers of all titles/subtitles of Section 2 (and correspondent titles of Tables 2a and 2b); actually, according to the comments of Referee 2 as well, we extensively revised Section 2 by changing the structure, splitting, adding, moving parts.

Reviewer 2 Report
Overall Comments:
In this manuscript “Molecular tools to identify and characterize malignant catarrhal fever viruses (MCFV) of ruminants and captive artiodactyla”, Bianchessi and co-authors try to summarize the molecular tools to identify and characterize the novel Malignant Catarrhal Fever (MCF) strains, intent to highlight advantages and drawbacks and to identify the gaps need to be filled in this field. This article has several obvious defects which cause it is not worth publication.
First of all, the overall structure of the article is simple, its structure is not an outline of a systematic review. This paper only simply summarizes the previous detection methods, lacks of a summary of relevant research progress in recent years (especially after 2006). In addition, the authors did not point out the use of mature and promising new methods and techniques in other similar viruses that have not been used in MCFV but have application prospects to supplement and improve the current detection methods. Furthermore, there are so many confusing places that make it difficult to follow up. Therefore, to the reviewer’s knowledge, the current status of the article does not meet the quality requirements for publication in viruses. This manuscript can be improved if authors follow the comment listed below.
Major issues:
1) Too many keywords are used, delete the unnecessary keywords, for example, delete evolution variants, degenerate primers, combine “malignant catarrhal fever; MCFV” as “malignant catarrhal fever (MCFV)”; summarize the three PCR method.
2) Introduction section: This section should be extensively revised. It should provide a systematic summary of relevant detection methods for MCFV, point out the challenges and shortcomings of the current detection platform, and highlight the novelty of this study. In addition, this section should be streamlined and condensed by combining some paragraphs and removing unnecessary redundancies. Some paragraphs should be the contents of an independent subtitle.
3) Also in the introduction section, the authors should provide a comprehensive and systematic summary of the research progress and detection methods in recent years.
4) References: Most of the cited references were published before 2006. Is there any new and significant updates of relevant detection methods in the past six years? The author should make a comprehensive summary and update.
Minor issues:
1) The subtitle is marked incorrectly and the title is repeated: For example, subtitle in Line 144 and Line 224 are repeated.
2) Line 285, check the title number.
Author Response
Point-by-point Response (in red) to Reviewer (in black)
Reviewer reports:
Reviewer #2: Comments and Suggestions for Authors:
Overall Comments:
In this manuscript “Molecular tools to identify and characterize malignant catarrhal fever viruses (MCFV) of ruminants and captive artiodactyla”, Bianchessi and co-authors try to summarize the molecular tools to identify and characterize the novel Malignant Catarrhal Fever (MCF) strains, intent to highlight advantages and drawbacks and to identify the gaps need to be filled in this field. This article has several obvious defects which cause it is not worth publication.
First of all, the overall structure of the article is simple, its structure is not an outline of a systematic review. This paper only simply summarizes the previous detection methods, lacks of a summary of relevant research progress in recent years (especially after 2006). In addition, the authors did not point out the use of mature and promising new methods and techniques in other similar viruses that have not been used in MCFV but have application prospects to supplement and improve the current detection methods. Furthermore, there are so many confusing places that make it difficult to follow up. Therefore, to the reviewer’s knowledge, the current status of the article does not meet the quality requirements for publication in viruses. This manuscript can be improved if authors follow the comment listed below.
We thank the reviewer for these comments and for the indication of possible ways to resolve the defects noted and to improve the manuscript. We agree with both reviewers that there is limited and old literature for the molecular analysis of this virus group; we extended the list of references in the present version of the manuscript; however, unfortunately, not many publications became available in this area recently. We extensively revised Section 2 by changing the structure, splitting, adding, moving parts.
We truly hope that the actual version of the manuscript, after the extensive and careful revisions, meets the high standard quality of the Journal.
Major Issues:
1) Too many keywords are used, delete the unnecessary keywords, for example, delete evolution variants, degenerate primers, combine “malignant catarrhal fever; MCFV” as “malignant catarrhal fever (MCFV)”; summarize the three PCR method.
We reduced the number of keywords according to the suggestions.
2) Introduction section: This section should be extensively revised. It should provide a systematic summary of relevant detection methods for MCFV, point out the challenges and shortcomings of the current detection platform, and highlight the novelty of this study. In addition, this section should be streamlined and condensed by combining some paragraphs and removing unnecessary redundancies. Some paragraphs should be the contents of an independent subtitle.
We extensively revised the Introduction section according to the suggestions. The present review manuscript was not aimed to be a systematic review, but rather more a structured review “State-of-the-science” type, based on a systematic methodology to identify the relevant literature on this methodological topic. We synthesized it, removed redundant or unnecessary details, moved few paragraphs, summarized the currently used methods and finally we subdivided the content of “previous” Introduction into two independent subparagraphs.
3) Also in the introduction section, the authors should provide a comprehensive and systematic summary of the research progress and detection methods in recent years.
According also with the other Referee’s comments, we implemented the manuscript by adding (in section 2) more recently developed methods.
4) References: Most of the cited references were published before 2006. Is there any new and significant updates of relevant detection methods in the past six years? The author should make a comprehensive summary and update.
We agree with the referee’s comments; we extended the list of references by adding the ones related to more recently developed methods; nevertheless, unfortunately, not many publications became available on this topic recently. The articles cited, albeit not many and somehow dated, are still the only available references for the methods applied to identify and characterize MCFV.
Minor Issues:
1) The subtitle is marked incorrectly and the title is repeated: For example, subtitle in Line 144 and Line 224 are repeated.
2) Line 285, check the title number.
In agreement with the comments, we reviewed the style, we corrected the titles and the numbers of all titles/subtitles of Section 2 (and correspondent titles of Tables 2a and 2b); actually, as above mentioned, we extensively revised Section 2 by changing the structure, splitting, adding, moving parts.

Round 2
Reviewer 2 Report
Overall Comments:
The authors reply all my comments. The manuscript is revised nicely. I recommend the paper should be accepted for publication after eliminate the format issues listed below.
Minor issues -
Section 2.1, it will be better if authors merged three paragraphs into one.
Line 181, Spaces should leave at the beginning of the first sentence.
Line 206, Please delete the extra row. This issue exists in many places, do this throughout the manuscript.
Author Response
Point-by-point Response (in red) to Reviewer (in black)
Reviewer report:
Reviewer #2: Comments and Suggestions for Authors:
Overall Comments:
The authors reply all my comments. The manuscript is revised nicely. I recommend the paper should be accepted for publication after eliminate the format issues listed below.
Minor issues -
Section 2.1, it will be better if authors merged three paragraphs into one.
As advised, we merged the previous three paragraphs of Section 2.1 into one.
Line 181, Spaces should leave at the beginning of the first sentence.
We modified the space according to the request at previous line the 181 (low line 179).
Line 206, Please delete the extra row. This issue exists in many places, do this throughout the manuscript.
We deleted the extra spaces at previous line 206 (now line 205), at line (now) 259 and in other places of the manuscript between sections.
Submission Date 14 October 2022
Date of this review 24 Nov 2022 04:26:05
